# Interactome analysis reveals that lncRNA HULC promotes aerobic glycolysis through LDHA and PKM2

Chunqing Wang[1,9], Yongmei Li[2,9], Shuai Yan[3,9], Hao Wang[3,4], Xianfeng Shao[3,7], Mingming Xiao[1], Baicai Yang[3,8], Guoxuan Qin[5], Ruirui Kong[6], Ruibing Chen [4✉] & Ning Zhang [1,6✉]

Interacting with proteins is a crucial way for long noncoding RNAs (lncRNAs) to exert their biological responses. Here we report a high throughput strategy to characterize lncRNA interacting proteins in vivo by combining tobramycin affinity purification and mass spectrometric analysis (TOBAP-MS). Using this method, we identify 140 candidate binding proteins for lncRNA highly upregulated in liver cancer (HULC). Intriguingly, HULC directly binds to two glycolytic enzymes, lactate dehydrogenase A (LDHA) and pyruvate kinase M2 (PKM2). Mechanistic study suggests that HULC functions as an adaptor molecule that enhances the binding of LDHA and PKM2 to fibroblast growth factor receptor type 1 (FGFR1), leading to elevated phosphorylation of these two enzymes and consequently promoting glycolysis. This study provides a convenient method to study lncRNA interactome in vivo and reveals a unique mechanism by which HULC promotes Warburg effect by orchestrating the enzymatic activities of glycolytic enzymes.

[1] Tianjin Medical University Cancer Institute and Hospital, National Clinical Research Center for Cancer, Key Laboratory of Cancer Prevention and Therapy, Tianjin's Clinical Research Center for Cancer, Tianjin Medical University, 300070 Tianjin, China. [2] Department of Pathogen Biology, School of Basic Medical Sciences, Tianjin Medical University, 300070 Tianjin, China. [3] Department of Genetics, School of Basic Medical Sciences, Tianjin Medical University, 300070 Tianjin, China. [4] School of Pharmaceutical Science and Technology, Tianjin University, 300072 Tianjin, China. [5] School of Microelectronics, Tianjin University, 300072 Tianjin, China. [6] Peking University First Hospital, Peking University Health Science Center, 100034 Beijing, China. [7] Present address: Tianjin Medical University Eye Hospital, Eye Institute & School of Optometry and Ophthalmology, 300384 Tianjin, China. [8] Present address: Department of Gynecology and Obstetrics, Jiaxing Maternity and Child Health Care Hospital, 314050 Jiaxing, China. [9] These authors contributed equally: Chunqing Wang, Yongmei Li, Shuai Yan. ✉email: rbchen@tju.edu.cn; zhangning@bjmu.edu.cn

Cellular metabolism reprogramming is a major hallmark of cancer[1]. To sustain tumor growth, cancer cells have an enhanced demand for nutrients to support dramatically elevated cell proliferation. Instead of using the tricarboxylic acid (TCA) cycle in mitochondria to generate ATP, cancer cells preferentially convert glucose to lactate through glycolysis, even in the presence of oxygen, which provides more metabolites for cell proliferation[2,3]. This aerobic glycolysis process plays important roles in tumorigenesis and tumor progression[4,5]. Multiple glycolytic enzymes have been reported to be upregulated in cancers and are associated with poor prognosis. For example, lactate dehydrogenase A (LDHA) is overexpressed in different cancers, and LDHA inhibition is shown to impair tumorigenesis and tumor growth[6,7]. Pyruvate kinase M2 (PKM2), that converts phosphoenolpyruvate to pyruvate, also plays a pivotal role in cancer progression[8]. However, it is still largely unclear how a cancer cell orchestrates the activities of PKM2 and LDHA to promote aerobic glycolysis and dampen the TCA cycle.

Long noncoding RNAs (lncRNAs) are defined as transcripts longer than 200 nt without an evident protein coding function[9]. The number of lncRNAs is estimated to range between less than 20,000 to over 100,000 in human cells[10]. These molecules play important regulatory roles in many biological processes, e.g., gene regulation, chromatin remodeling, and cell fate determination during development[11]. Dysregulation of lncRNAs has been observed in many different types of cancer, regulating a diverse set of important cancer hallmarks, e.g., proliferation[12], apoptosis[13], metastasis[14], metabolism alteration[15], and cancer-related inflammation[16]. LncRNA highly upregulated lncRNAs in liver cancer (HULC) is overexpressed in hepatocellular carcinoma (HCC) tissues[17]. The transcription of HULC is activated by transcription factor cAMP-responsive element-binding protein (CREB)[18], and the upregulation of HULC has been reported to promote HCC cell proliferation and invasion both in vitro and in vivo[19,20]. HULC could modulate gene expression through targeting miRNAs and mRNAs;[19,21] however, it seems that these observations cannot account for the profound roles of HULC in hepatoma progression. A comprehensive analysis of its interacting proteome is required to further understand its biological functions and the underlying mechanisms.

Recent studies have greatly expanded the known repertoire of RNA-binding proteins (RBPs), and many proteins without classical RNA-binding domains are identified as novel RBPs[22]. For example, some metabolic enzymes are found to bind with RNAs as a gene regulation mechanism[23]. It is speculated that RNA binding could also modulate the metabolic functions of these enzymes; however, more evidences are required to support this theory, highlighting the need for a more in-depth study of unconventional RNA-protein interactions.

Several high throughput technologies have been developed to study RNA and protein interactions[22]. Some of them are protein-centric methods based on next-generation sequencing, e.g., crosslinking-immunoprecipitation RNA-sequencing (CLIP-Seq)[24]. However, RNA-centric techniques are often required to explore the unknown functions of lncRNAs. RNA pull-down is the most widely used method to identify lncRNA interacting proteins in vitro[25], but it cannot distinguish the interactions that occur in vivo from those that occur in a solution. Several methods have been developed to study RNA-binding proteins in vivo, e.g., PNA-assisted identification of RNA-binding proteins (PAIR)[26], chromatin isolation by RNA purification[27], capture hybridization analysis of RNA targets (CHART)[28], RNA antisense purification (RAP)[29], and MS2 biotin-tagged RNA affinity purification (MS2-BioTRAP)[30]. PAIR employs synthesized peptide-nucleic acid probes: the peptide allows cell penetration and the oligonucleotide is used to hybridize with the target RNA[26]. Although the

probes are nicely designed, this method suffers from high cost and low detection sensitivity. ChIRP, CHART, and RAP all use antisense oligonucleotides to isolate the endogenous lncRNA of interest. These methods usually require large input cell numbers and may not be competent for lncRNA molecules with low abundances, and they may also suffer from false positives induced by interference from other RNAs containing homolog sequences[22]. MS2-BioTRAP uses four MS2 stem loops and bacteriophage MS2 coat protein to isolate the target RNA[30]. Hartmuth et al. developed a tobramycin affinity-selection method to study the composition of spliceosomes using J6f1 aptamer (40 nt) that binds with the small molecule tobramycin with high affinity[31]. Compared to the MS2-BioTRAP system, J6f1 is shorter and does not require the expression of a foreign protein, and it may provide an alternative and more straightforward approach to identify lncRNA-interacting proteins in vivo.

Here, we establish a high throughput strategy to characterize the interacting proteome of lncRNA-HULC by combining tobramycin affinity purification and quantitative mass spectrometry analysis (TOBAP-MS). Using this method, we identify 140 potential HULC interacting proteins and build a highly connected interactome network. Intriguingly, the results show that HULC interacts with two glycolytic enzymes, LDHA and PKM2. Mechanistic study validates these interactions and reveals that HULC-promoted aerobic glycolysis by directly binding to LDHA and PKM2 and modulating their enzymatic activities.

## Results

**The identification of lncRNA-HULC interacting proteins.** In this study, we established a TOBAP-MS method to isolate and identify the interacting proteins of lncRNA-HULC in vivo. First, a 40 nt RNA aptamer J6f1 (Fig. 1a) that binds to tobramycin with high affinity ($K_d = 5$ nM) was engineered to the 3′ end of the full-length sequence of *HULC* to construct the J6f1-tagged *HULC* vector. The binding efficiency of HULC-J6f1 RNA to tobramycin was tested in vitro. The results showed that the synthesized HULC-J6f1 bound efficiently to tobramycin-derivatized Sepharose beads (Fig. 1b). Moreover, the HULC-J6f1-encoding vector could be successfully transfected into HepG2 cells with similar expression level as the wild-type HULC (Fig. 1c).

Next, we examined whether the J6f1 tag would affect the structure and the biological functions of HULC. MFold (http://unafold.rna.albany.edu/) was employed to analyze the secondary structures of HULC-J6f1 and HULC. The results indicated that J6f1 forms a hair-pin structure at the 3′ end without interfering with the rest of the RNA strand (Supplementary Fig. 1). In addition, overexpression of HULC-J6f1 had similar effect on cell proliferation compared with the wild-type HULC (Fig. 1c, d). Moreover, the cellular localizations of HULC and HULC-J6f1 were imaged by RNA fluorescence in situ hybridization (FISH) using biotinylated antisense probes, showing similar distribution of HULC and HULC-J6f1 in HepG2 cells (Fig. 1e). Taken together, these data indicate that the J6f1 aptamer could be used to efficiently purify the lncRNA of interest without interfering with its biological behaviors.

The quantitative proteomics workflow is shown in Fig. 1f. HepG2 cells transfected with the J6f1 vector were labeled using SILAC medium containing both $^{13}C_6$ lysine and $^{13}C_6$ arginine, and the HepG2-HULC-J6f1 cells were cultured in normal cell medium. The cell lysates were incubated with tobramycin-conjugated beads to isolate HULC-J6f1 and its interacting proteins. Next, the beads were mixed, and the eluted proteins were separated using SDS-PAGE. Each gel lane was divided into ten bands and then digested with trypsin for downstream

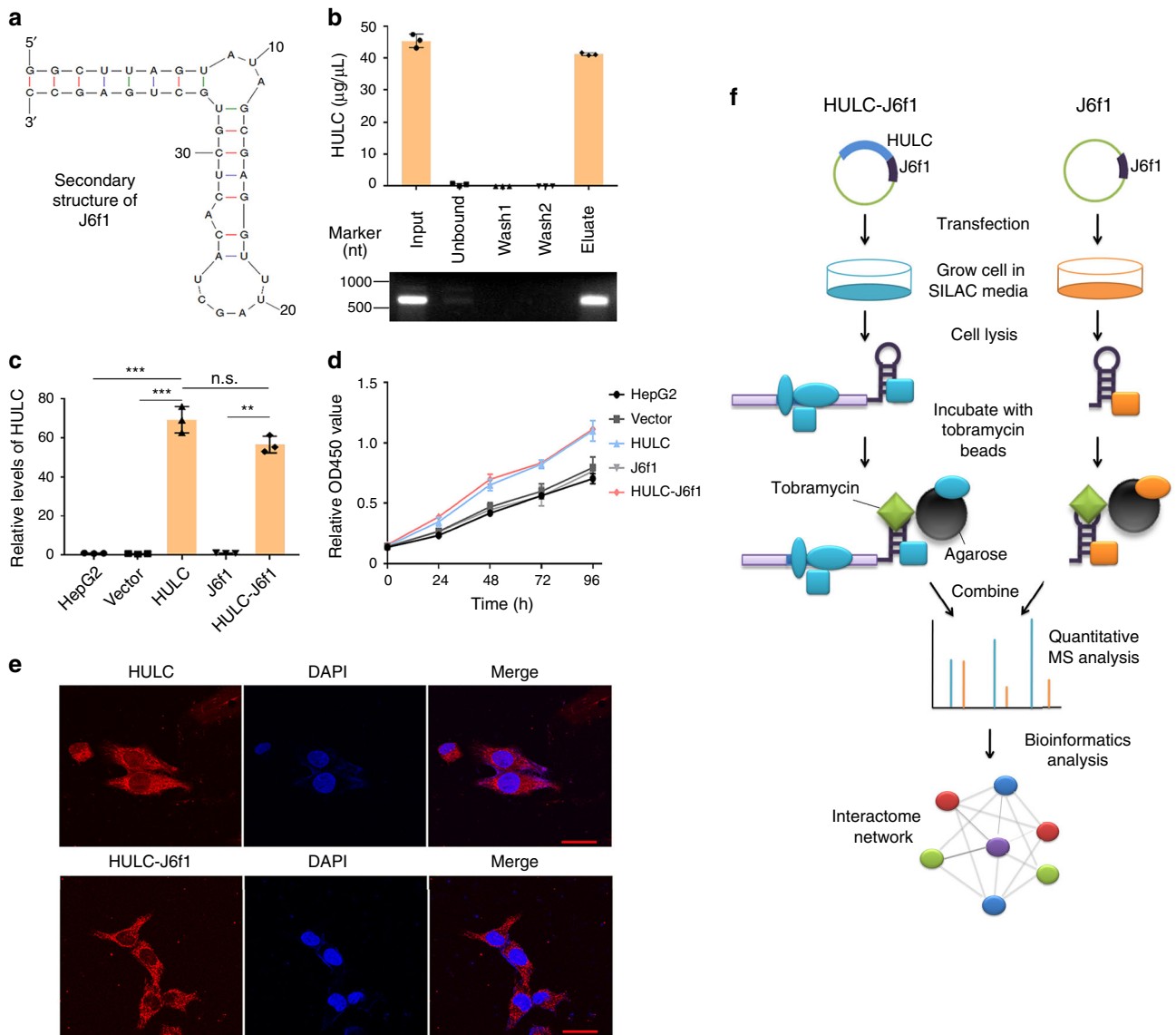

**Fig. 1 The mass spectrometric strategy for characterizing HULC interacting proteins. a** The secondary structure of the J6f1 aptamer predicted by MFold (http://unafold.rna.albany.edu/). **b** In vitro binding of HULC-J6f1 to tobramycin-conjugated agarose beads. HULC-J6f1 was synthesized by in vitro transcription. Data represent mean ± s.d. of triplicate independent analyses. **c** The levels of HULC in HepG2 cells overexpressing HULC or HULC-J6f1 were measured by qRT-PCR. HepG2 cells expressing empty vector or J6f1 vector were used as controls. Data represent mean ± s.d. ($n = 3$ independent experiments, ***$P < 0.001$, n.s., not significant, by two-sided Student's $t$ test). **d** The proliferation of HepG2 cells expressing HULC or HULC-J6f1 was measured by CCK8 assay at the indicated time points. Data represent mean ± s.d. of triplicate independent experiments. **e** The cellular localizations of HULC and HULC-J6f1 were analyzed by RNA-FISH. The scale bar was 20 μm. **f** The workflow of tobramycin affinity purification mass spectrometry (TOBAP-MS). See also Supplementary Fig. 1. Source data are provided as a Source data file.

LC–MS/MS analyses. Finally, bioinformatics analysis was performed to construct the interactome network of HULC.

**Interactome network shows the biological roles of HULC.** The isolated HULC interacting proteins were analyzed with an LTQ-Orbitrap Fusion mass spectrometer. As a result, 140 proteins were detected with log$_2$ ratio $_{HULC /control} \geq 1.5$ and $P$ value $\leq 0.05$ in the SILAC quantification experiment (Fig. 2a and Supplementary Data 1). To further understand the potential biological functions of HULC, pathway and gene ontology analysis was performed on the identified proteins (Supplementary Fig. 2). KEGG database analysis indicated that HULC interacting proteins were significantly associated pathways involved in virus response, glycolysis, etc. Biological process analysis showed that the identified proteins were enriched in response to virus, cell–cell adhesion,

RNA splicing, etc. Protein class analysis revealed that the majority of the identified proteins were related with RNA binding and protein binding.

To build the protein interactome network of HULC, the identified proteins were searched against the STRING protein–protein interaction (PPI) database for known PPIs, and the protein nodes were grouped based on their known functions acquired through GO analysis. As a result, a highly connected network comprising 67 proteins and 190 connections was mapped (Fig. 2b). These proteins can be divided into seven major sub-clusters based on their known biological functions, including metabolism, cell adhesion, response to virus, etc. The protein interactome analysis provides a systematic view of the biological functions of HULC, showing its potential roles in diverse biological processes.

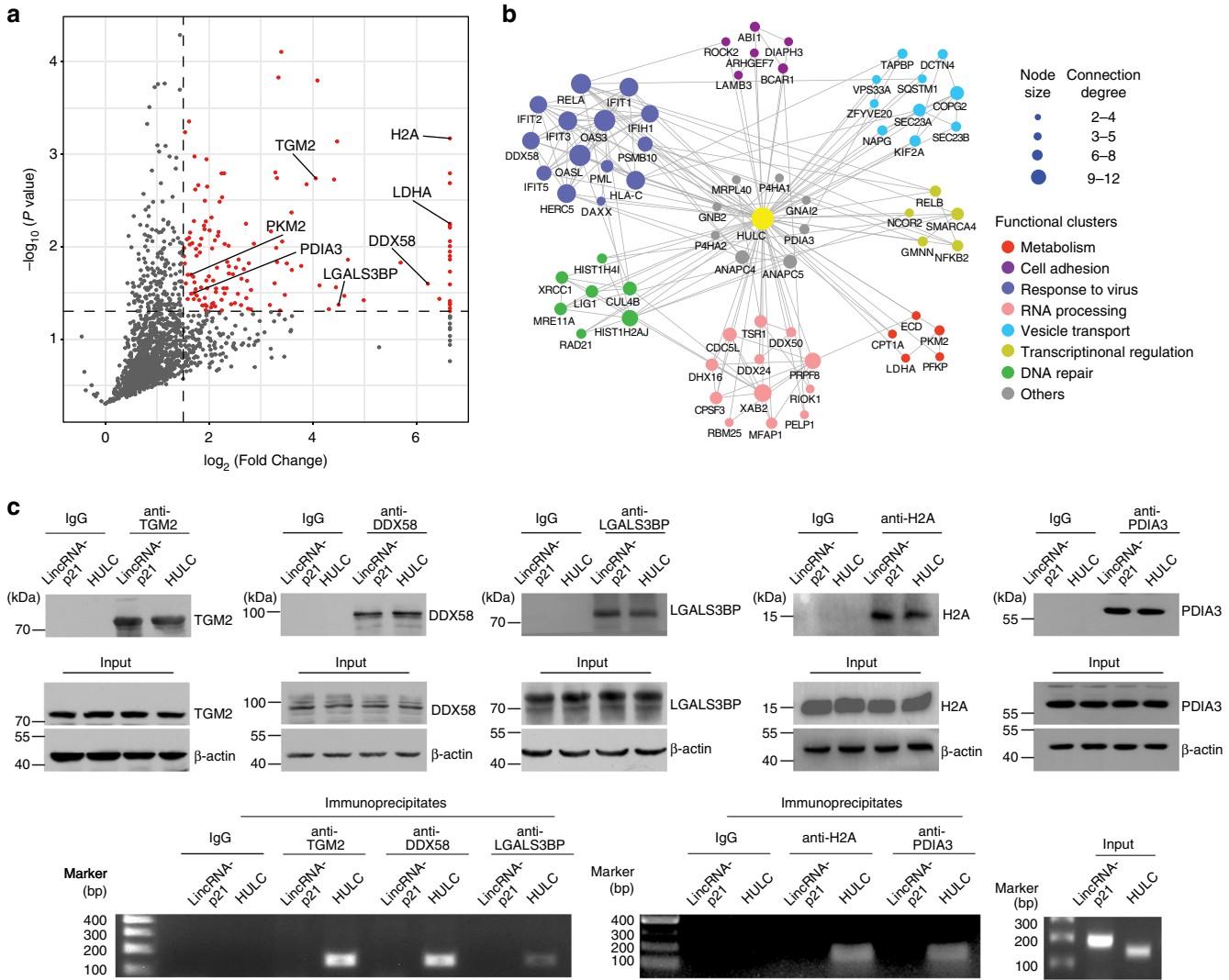

**Fig. 2 The identified HULC interacting proteins. a** The volcano plot of the identified proteins. The proteins significantly enriched in the tobramycin purified samples are shown as red dots. Log$_2$ fold change was plotted on the x-axis and $-\log_{10} P$ value was plotted on the y-axis. **b** The protein interactome network of HULC. PPI information was obtained through a database search using String 9.0 and imported into Cytoscape 3.1.1 for network construction. Proteins and their interactions are shown as nodes and edges. Node size reflects the interaction degree. Proteins are grouped based on their known biological functions. **c** Validation of selected identified proteins by RIP. Immunoblots of TGM2, DDX58, LGALS3BP, H2A and PDIA3 in the cell lysates and immunoprecipitates are shown in the upper panel. The agarose gel electrophoresis images of HULC amplified by qRT-PCR are shown in the lower panel. LincRNA-p21 was used as an irrelevant lncRNA control. See also Supplementary Figs. 2 and 3. Source data are provided as a Source Data file.

Furthermore, we conducted a RAP-MS experiment using antisense DNA probes for purification[29]. Endogenous HULC-binding proteins were purified by seven manually designed 50-nt antisense DNA probes and identified by quantitative mass spectrometry. As a result, 25 proteins were detected as potential HULC-binding proteins (Supplementary Table 1 and 2), and four of them were also observed by TOBAP-MS, including LDHA, protein disulfide-isomerase A3 (PDIA3), histone H2A, and protein-glutamine gamma-glutamyltransferase 2 (TGM2). Although more cells were used in the RAP-MS experiment, much fewer proteins were detected. In addition, about half of these proteins were detected with only two unique peptides, which are normally considered as low-confident identifications. The lower coverage of RAP-MS may due to several possible reason, e.g. the low endogenous level of HULC in the cell, low efficiency of UV crosslinking, and inaccessibility of target sequences in RAP. In addition, RAP only captures direct binding proteins cross-linked to the target RNA by UV irradiation. In

contrast, in the TOBAP procedure, the target RNA is over-expressed, and this method isolates both direct and indirect binding proteins. The data suggest that these two methods may provide complementary information, and TOBAP-MS recovers more RNA-binding protein candidates compared to RAP-MS.

To evaluate the identification results, we examined three common proteins identified by both methods (TGM2, H2A and PDIA3) and another two randomly selected proteins specifically observed by TOBAP-MS (DDX58 and LGALS3BP). Their interactions with HULC were validated by RNA immunopreci-pitation (RIP), followed by qRT-PCR and agarose gel analysis (Fig. 2c). We observed enrichment of HULC in the immunopre-cipitates of all the five proteins. A cytoplasm-localized lncRNA lincRNA-p21 was examined as an irrelevant RNA control to evaluate the specificity of the identified interactions. As a result, lincRNA-p21 was not detected in the immunoprecipitates of the selected proteins, which suggested that the identified interactions were specific for HULC. In addition, no enrichment of HULC was

detected in the immunoprecipitates of two irrelevant proteins, including glyceraldehyde 3-phosphate dehydrogenase (GAPDH) and the mammalian target of rapamycin (mTOR), further demonstrating the specificity of the identification (Supplementary Fig. 3). These results indicate that the TOBAP-MS method is effective for identifying lncRNA interacting proteins.

**HULC binds glycolytic enzyme LDHA and elevates its activity**. Intriguingly, TOBAP-MS analysis revealed that HULC may interact with several enzymes from the glycolysis pathway (Fig. 2b), indicating a potential role of HULC in the regulation of glucose metabolism. It has been well documented that LDHA and PKM2 are overexpressed in a number of cancers and play pivotal roles in the Warburg effects[5]. Thus, we focused on the interactions between HULC and these two glycolytic enzymes in the following study.

First, we explored the interaction between HULC and LDHA. Lactate dehydrogenase (LDH) catalyzes the reversible conversion of pyruvate to lactate, which is the last step of glycolysis, and there are five isozymes of LDH yielded by different combinations of two types of subunits, LDHA and LDHB[32]. Evidences have shown that LDHA is normally overexpressed in cancers and promotes tumor progression[6,7], but the role of LDHB is rather controversial and may be dependent on the tissue type[33,34]. Our TOBAP-MS analysis revealed that HULC potentially interacted with LDHA (Fig. 2a). Results from RIP experiment showed that HULC was enriched in LDHA immunoprecipitates, which further confirmed the interaction between HULC and LDHA (Fig. 3a). On the other hand, the control lincRNA-p21 was not detected in the immunoprecipitates of LDHA, suggesting that the HULC/LDHA interaction was specific (Fig. 3a). Next, we examined the co-localization of HULC and LDHA by RNA-FISH combined with immunofluorescence. The results showed a cytoplasmic co-localization pattern between these two molecules (Fig. 3b).

Next, we sought to understand whether HULC could directly bind to LDHA. RNA pull-down assay was performed by using biotinylated HULC and antisense HULC. Western blotting of the pull down showed that LDHA, but not LDHB, bound to HULC (Fig. 3c and Supplementary Fig. 4). To further determine whether HULC could directly bind to LDHA, an in vitro His-tag pull-down assay was performed by using recombinant proteins and in vitro transcribed HULC. The results showed that rLDHA, instead of rLDHB, bound to HULC in vitro (Fig. 3d). Moreover, the molecular interaction between HULC and LDHA was analyzed by surface plasmon resonance (SPR), and the dissociation constant $K_d$ between these two molecules was determined to be $2.898 \times 10^{-8}$ M (Fig. 3e). Taken together, the data indicate that HULC directly binds to LDHA.

Furthermore, we studied the consequences of the HULC/LDHA binding. The knockdown of HULC by different shRNA sequences reduced LDH activity, and the overexpression of HULC enhanced LDH activity (Fig. 3f, g). It has been reported that LDHA activity is regulated by its phosphorylation;[35] thus, we speculated that HULC might regulate LDHA phosphorylation. As shown in Fig. 3h, the knockdown of HULC reduced the Y10 phosphorylation on LDHA, and the phosphorylation level of LDHA was enhanced in cells with HULC overexpression. These results indicate that HULC regulates LDH enzymatic activities potentially through modulating its phosphorylation.

**HULC binds PKM2 and dampens its enzymatic activity**. PKM2 plays a pivotal role in cancer cell metabolism[8]. Our TOBAP-MS experiment suggested an interaction between HULC and PKM2 (Fig. 2a). This interaction was further confirmed with an RNA

pull-down assay. PKM2 was pulled down from HepG2 cell lysates by in vitro transcribed biotinylated HULC instead of antisense HULC (Fig. 4a). RNA-FISH combined with immunofluoresence showed that PKM2 and HULC co-localized in the cytoplasm (Fig. 4b). The *PKM* gene encodes two alternatively spliced transcripts, i.e., PKM1 and PKM2[36]. PKM1, with high constitutive enzymatic activity, is generally expressed in normal tissues. In contrast, PKM2 is less active and promotes aerobic glycolysis and tumor growth[8]. In the RIP assay, HULC was enriched with both anti-PKM1 and anti-PKM2 antibodies, but with higher intensity in the PKM2 immunoprecipitates (Fig. 4c). The sequences of PKM1 and PKM2 are different by only one exon, which might contribute to the HULC-binding specificity of PKM2. To test this hypothesis, flag-tagged exon 9 and exon 10 of the *PKM* gene were expressed in HepG2 cells, and their interactions with HULC were examined by RIP. The results showed that HULC only co-precipitated with the protein product of exon 10, the specific exon for PKM2 (Fig. 4d).

Furthermore, we studied whether the interaction between PKM2 and HULC was also direct. First, in vitro His-tag pull-down assay showed that rPKM2 could directly bind to HULC instead of the antisense HULC (Fig. 4e). In addition, the molecular interaction between HULC and PKM2 was analyzed, and the dissociation constant $K_d$ between these two molecules was determined to be $2.045 \times 10^{-7}$ M (Fig. 4f). The results demonstrate that HULC directly binds to glycolytic enzyme PKM2.

We speculated that the binding of HULC to PKM2 may modulate the enzymatic activity of the latter. Indeed, PK activity was increased in HULC knockdown cells, and the overexpression of HULC decreased PK activity (Fig. 4g). PKM2 fluctuates between two major states: an active tetrameric form and a less active dimeric or monomeric form[37]. Thus, we examined whether HULC could alter the tetramer formation of PKM2. The upregulation of HULC did not affect the overall expression of PKM2, but in vivo crosslinking assay revealed that the overexpression of HULC led to a marked decrease in the formation of tetrameric PKM2 (Fig. 4h). Since the tetramer formation of PKM2 is affected by its posttranslational modifications, such as phosphorylation[38], we next examined the phosphorylation level of PKM2 under different conditions. Intriguingly, the knockdown of HULC reduced the Y105 phosphorylation of PKM2, and the upregulation of HULC enhanced its phosphorylation (Fig. 4i). Collectively, the data indicate that HULC directly binds with PKM2 and downregulates its enzymatic activity by promoting PKM2 phosphorylation and inhibiting its tetramer formation.

**HULC modulates LDHA and PKM2 through FGFR1**. We first tested whether HULC might directly link LDHA and PKM2. However, we did not observe any effect of HULC on the interaction between LDHA and PKM2 in vitro or in vivo. It has been reported that the phosphorylations of LDHA and PKM2 are both regulated by fibroblast growth factor receptor type 1 (FGFR1), a membrane receptor tyrosine kinase, which is observed to be overexpressed in many types of cancer[39]. Direct phosphorylation of LDHA at Y10 and Y83 by FGFR1 could enhance its enzymatic activity[34,35]. Meanwhile, FGFR1 phosphorylates multiple tyrosine sties of PKM2, and the phosphorylation of Y105 inhibits its tetramer formation and reduces its enzymatic activity[38]. Therefore, we suspected that HULC might regulate the FGFR1-mediated phosphorylation of LDHA and PKM2.

We first investigated whether HULC could also bind with FGFR1. The RIP experiment indicated that HULC was enriched in the immunoprecipitates of FGFR1 by ~100-fold compared with that of the normal IgG control (Fig. 5a). Moreover, the

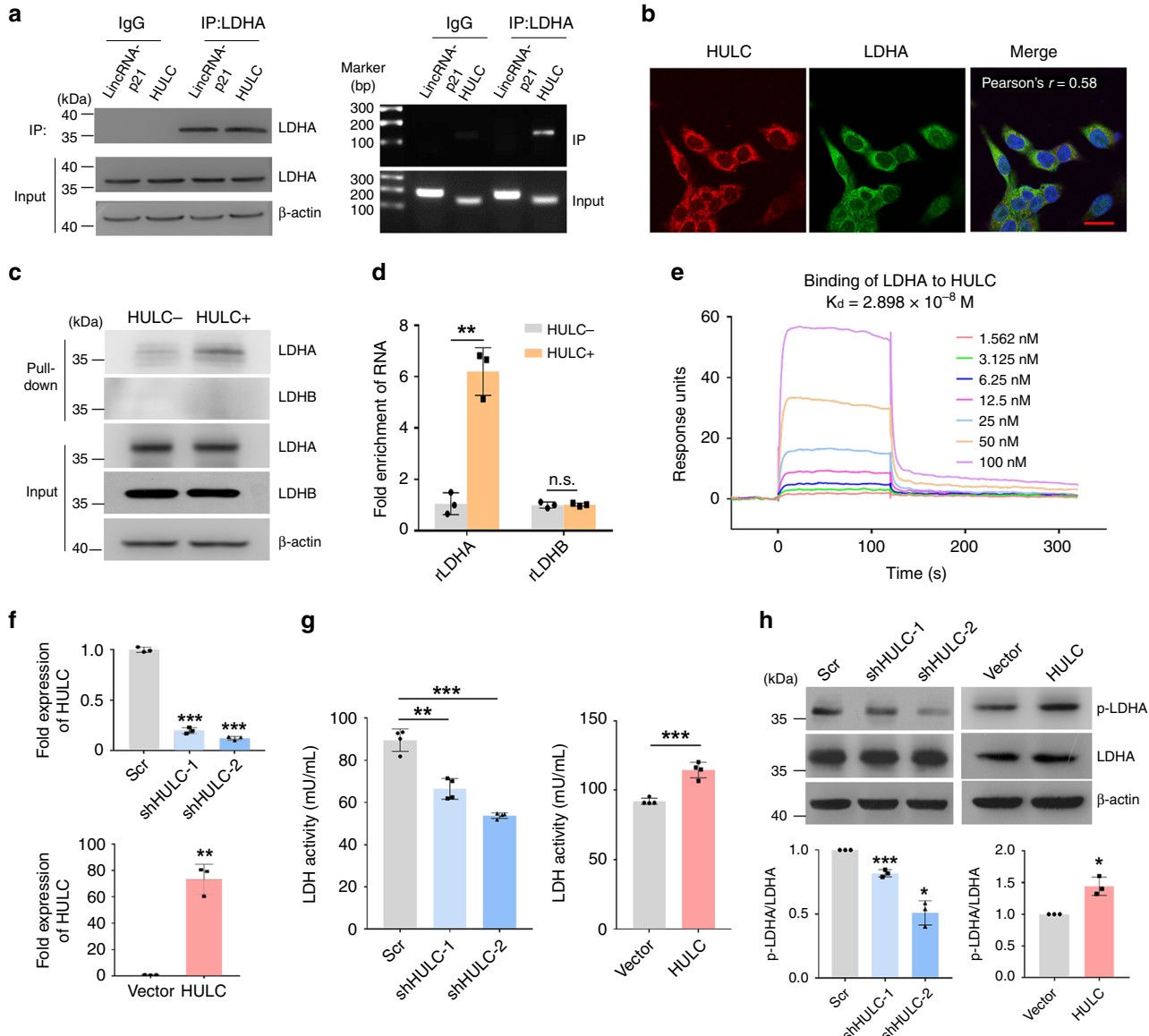

**Fig. 3 HULC interacts with the glycolytic enzyme LDHA. a** Validation of the interaction between LDHA and HULC. Immunoblots of LDHA in the cell lysates and immunoprecipitates of LDHA are shown in the left panel. Agarose gel electrophoresis images of HULC amplified by qRT-PCR are shown in the right panel. LincRNA-p21 was examined as the RNA control. **b** The cellular localizations of HULC and LDHA were analyzed by combining RNA-FISH and immunofluorescence. Cell nuclei were stained with DAPI, and the scale bar was 20 μm. **c** Biotinylated HULC was incubated with HepG2 cell lysate and then isolated by streptavidin-conjugated beads. LDHA and LDHB in the cell lysate and RNA pull-down were examined by western blotting. Biotinylated antisense HULC was used as the control. **d** His-tagged rLDHA or rLDHB was incubated with Dynabeads® His-tag isolation magnetic beads, respectively. Next, in vitro transcribed HULC or antisense HULC was incubated with the beads. Then, the RNA-protein complexes were isolated, and the levels of HULC were examined using qRT-PCR. Data represent the mean ± s.d. of triplicate independent experiments (***$P < 0.001$, n.s., not significant, by two-sided Student's $t$ test). **e** The molecular interaction between LDHA and HULC was analyzed to measure the dissociation factor $K_d$. **f** The levels of HULC in HepG2 cells with HULC knockdown (upper panel) or overexpression (lower panel) were measured by qRT-PCR. Data represent the mean ± s.d. of triplicate independent experiments (***$P < 0.001$, by two-sided Student's $t$ test) **g** The LDH activities of HepG2 cells with HULC knockdown (left panel) or overexpression (right panel) were examined by a lactate dehydrogenase activity assay kit and compared with the same number of corresponding control cells. Data represent the mean ± s.d. ($n = 4$ independent experiments, ***$P < 0.001$, by two-sided Student's $t$ test). **h** HULC was silenced or overexpressed in HepG2 cells. The levels of p-LDHA ($Y^{10}$) and LDHA in the cell lysates were detected by western blotting. Band intensities were measured by ImageJ. Data represent the mean ± s.d. of triplicate independent experiments (*$P < 0.05$, ***$P < 0.001$, by two-sided Student's $t$ test). See also Supplementary Fig. 4. Source data are provided as a Source data file.

overexpression of HULC promoted the phosphorylation of LDHA (Y10) and PKM2 (Y105), which were impaired by the treatment with the FGFR1 inhibitor PD166866 in a dose-dependent manner, suggesting that HULC indeed modulated the phosphorylation of these two enzymes through FGFR1 (Fig. 5b). Furthermore, the results from RNA pull-down assay

indicated that in vitro transcribed HULC could pull down the recombinant cytoplasmic fragment of FGFR1 (AA399-822) and rLDHA together, and similar results were observed for rPKM2 (Fig. 5c). Next, we examined whether the binding of HULC could affect the interactions between FGFR1 and these two enzymes. The His-tag pull-down assay revealed that rFGFR1 (AA399-822)

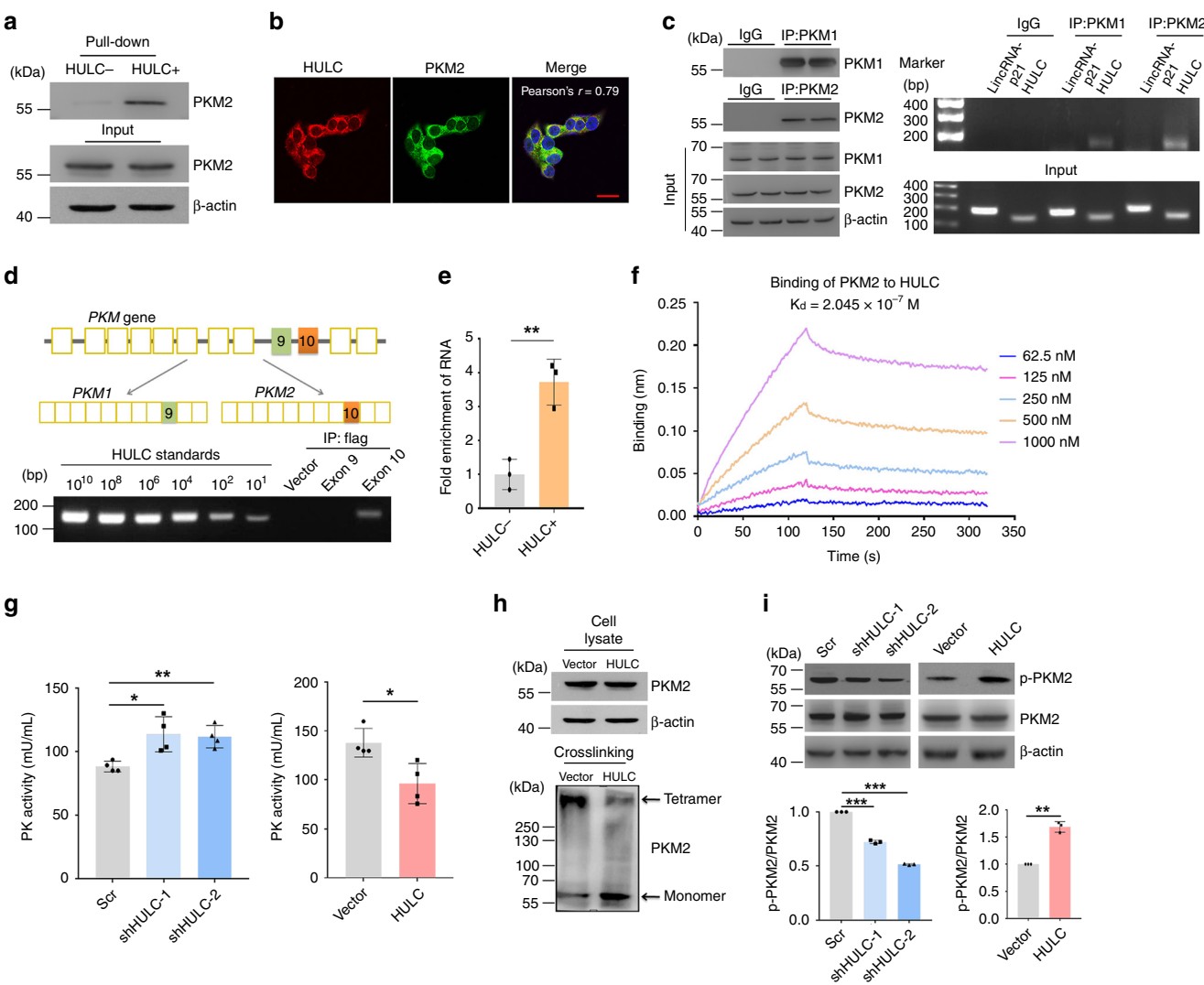

**Fig. 4 HULC interacts with the glycolytic enzyme PKM2. a** Biotinylated HULC and antisense HULC were synthesized by in vitro transcription and incubated with HepG2 cell lysates, respectively. The RNA-protein complexes were isolated with streptavidin-conjugated beads. PKM2 in the pull down was examined by western blotting. Biotinylated antisense HULC was used as the control. **b** The cellular localizations of HULC and PKM2 were analyzed by RNA-FISH combined with immunofluorescence. Cell nuclei were stained with DAPI, and the scale bar was 20 μm. **c** Immunoprecipitation of PKM1 and PKM2. The left panel shows the immunoblots of PKM1 and PKM2 in the cell lysate and immunoprecipitates. The right panel shows the agarose gel electrophoresis images of the qRT-PCR products of HULC. LincRNA-p21 was examined as RNA control. **d** Binding of HULC to flag-tagged exon 9 (PKM1 specific) and exon 10 (PKM2 specific) as determined by the RIP assay. **e** His-tagged rPKM2 was first immobilized to Dynabeads® His-tag isolation magnetic beads, and then incubated with in vitro transcribed HULC or antisense HULC. The RNA-protein complexes were isolated, and the levels of HULC were examined by qRT-PCR. Data represent the mean ± s.d. of triplicate independent experiments (***$P < 0.001$, by two-sided Student's $t$ test). **f** The molecular interaction between PKM2 and HULC was to measure the $K_d$ value. **g** The PK activities of HepG2 cells with HULC knockdown (left panel) or overexpression (right panel) were measured by a PK activity assay kit and compared with the same number of corresponding control cells. Data represent the mean ± s.d. ($n = 4$ independent experiments, *$P < 0.05$, by two-sided Student's $t$ test). **h** Western blotting of PKM2 in cells with or without crosslinking. In vivo crosslinking was performed by incubating the cells with 1 mM disuccinimidyl suberate (DSS) at room temperature for 30 min. **i** HULC was knocked down or overexpressed in HepG2 cells. The levels of p-PKM2 (Y[105]) and PKM2 in the cell lysates were detected by western blotting. Band intensities were measured by ImageJ. Data represent the mean ± s.d. of triplicate independent experiments (**$P < 0.001$, ***$P < 0.001$, by two-sided Student's $t$ test). Source data are provided as a Source data file.

directly binds with rLDHA, and the addition of HULC instead of the antisense HULC increased the interaction between rFGFR1 (AA399-822) and rLDHA (Fig. 5d). Similar results were observed for rPKM2 (Fig. 5d).

Since FGFR1 is a membrane protein, we suspected that HULC may affect the cellular localizations of LDHA and PKM2. Indeed, western blotting of the subcellular fractions showed that overexpression of HULC increased the membrane localizations of both LDHA and PKM2 (Fig. 5e). Next, we further investigated

the co-localizations of the studied molecules by RNA FISH combined with immunofluorescence. The results showed that stimulation by FGF induced the trans-localization of LDHA, PKM2 and HULC to the cell membrane, where they co-localized with FGFR1 (Fig. 5f).

Finally, we investigated which FGFR1-targeted phosphorylation sites on LDHA and PKM2 were responsible for the phenotype of HULC using phosphorylation mutants. The endogenous expression of these two genes was knocked down

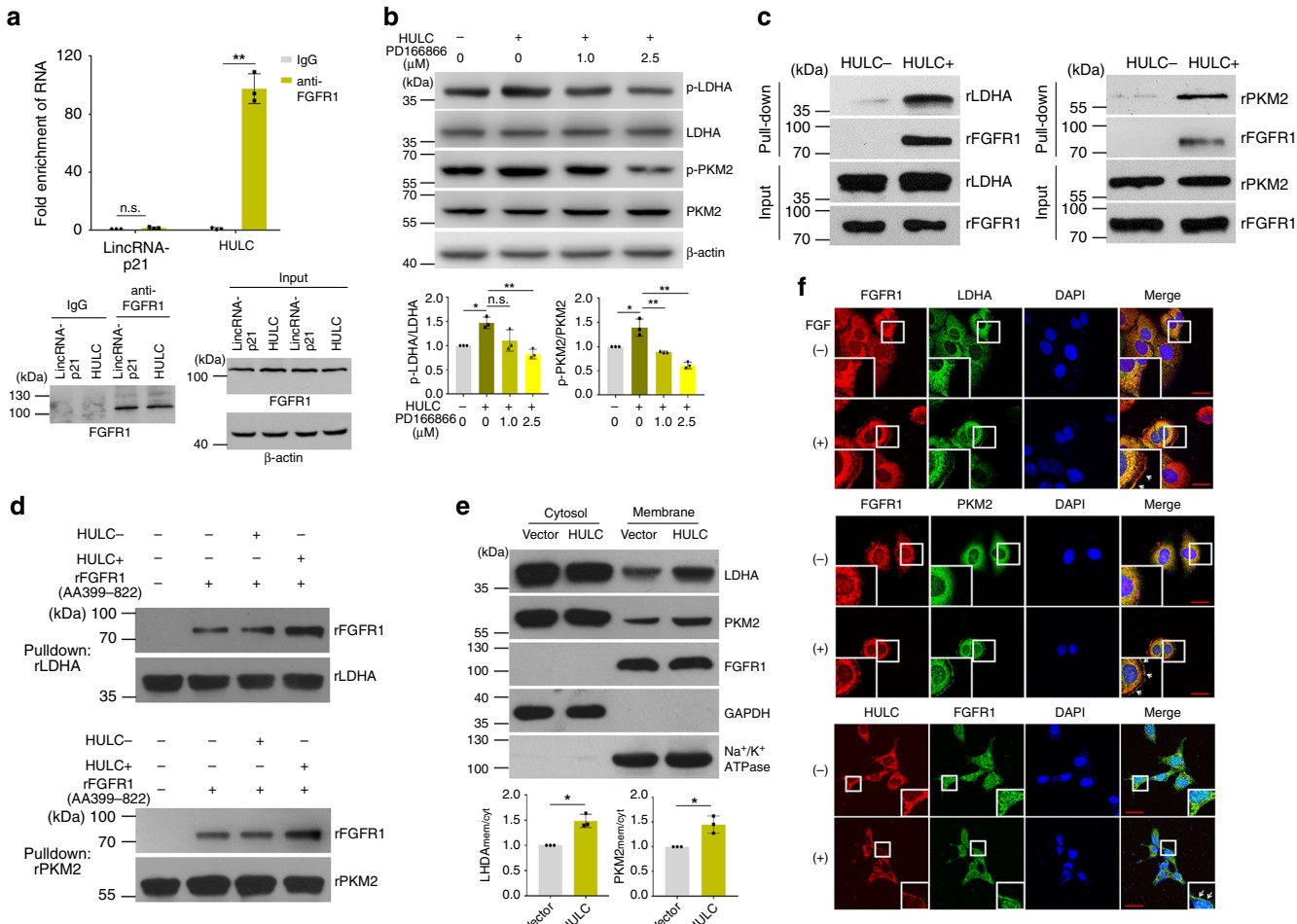

**Fig. 5 HULC modulates the phosphorylation of LDHA and PKM2 through FGFR1. a** qRT-PCR analysis of the enrichment of HULC in FGFR1 immunoprecipitates. LincRNA-p21 was used as the RNA control. Data represent mean ± s.d. of triplicate independent analyses (***$P < 0.001$, n.s., not significant, by two-sided Student's $t$ test). Immunoblots of FGFR1 in the cell lysates and immunoprecipitates are shown in the lower panel. **b** Immunoblots of p-LDHA ($Y^{10}$), LDHA, p-PKM2 ($Y^{105}$), and PKM2 in the cell lysates. HepG2 cells overexpressing HULC were incubated with the indicated concentrations of FGFR1 inhibitor PD166866 for 24 h. DMSO was added instead in the control groups. Band intensities were measured by ImageJ. Data represent mean ± s.d. of triplicate independent experiments. **c** In vitro transcribed biotinylated HULC was incubated with rFGFR1 and rLDHA (left panel) or rPKM2 (right panel) at 4 °C for 3 h, and the RNA-protein complexes were isolated with streptavidin-conjugated beads. Levels of rLDHA or rPKM2 in the pull down were examined by western blotting. Biotinylated antisense HULC was used as control. **d** His-tag pull-down assay results showing that HULC enhances the binding between FGFR1 and the two glycolytic enzymes. His-tagged rLDHA (upper panel) or rPKM2 (lower panel) was incubated with Dynabeads® His-tag isolation magnetic beads at 4 °C for 4 h. Next, the beads were isolated and then incubated with in vitro transcribed HULC or antisense HULC, followed by the addition of rFGFR1 (AA399–822). Finally, the beads were isolated, and rLDHA and rFGFR1 in the pull down were examined by western blotting. **e** Immunoblots of LDHA, PKM2, and FGFR1 in the cytosol and membrane fractions. GAPDH was examined as the cytosol marker, and Na+/K+ ATPase was used as the membrane marker. Band intensities were measured by ImageJ. Data represent mean ± s.d. of triplicate independent experiments (*$P < 0.05$, by two-sided Student's $t$ test). **f** Representative micrographs of FGFR1, LDHA, PKM2, and HULC in cells treated with or without 10 ng mL$^{-1}$ FGF. Cell nuclei were stained with DAPI, and the scale bar was 20 μm. See also Supplementary Fig. 5. Source data are provided as a Source data file.

from the cells followed by rescue with empty vector, wild type, or phosphorylation site-mutated LDHA or PKM2, respectively. Cell counting kit-8 (CCK8) assay showed that upregulation of HULC increased cell proliferation, and knockdown of LDHA or PKM2 reduced cell proliferation to a similar level as the control cells (Supplementary Fig. 5a–f). Overexpressing HULC did not promote cell proliferation in LDHA knockdown cells rescued with empty vector (Supplementary Fig. 5g–i). In cells rescued with LDHA-WT, overexpression of HULC enhanced cell proliferation, but overexpressing LDHA-Y10F or LDHA-Y83F mutants failed to rescue the HULC-promoted cell proliferation (Supplementary Fig. 5g–i). The results suggest that the Y10 and Y83 phosphorylation sites of LDHA play an important role in

mediating the effects of HULC. PKM2-Y105F could also block the effect of HULC on cell proliferation, and PKM2-Y83F and PKM2-Y370F showed similar effect as PKM2-WT (Supplementary Fig. 5j–l). The results agree with the previous report that FGFR1 phosphorylates multiple tyrosine sites on PKM2, but the Y83F and Y370F mutants have no significant effect on the enzymatic activity of PKM2[38]. These data indicate that the phosphorylation of LDHA at Y10 and Y83, and PKM2 at Y105 play an important role in mediating HULC-elicited cell proliferation.

Taken together, the results indicate that HULC may function as an adapter molecule that enhances the interactions between FGFR1 and these two glycolytic enzymes, increases their

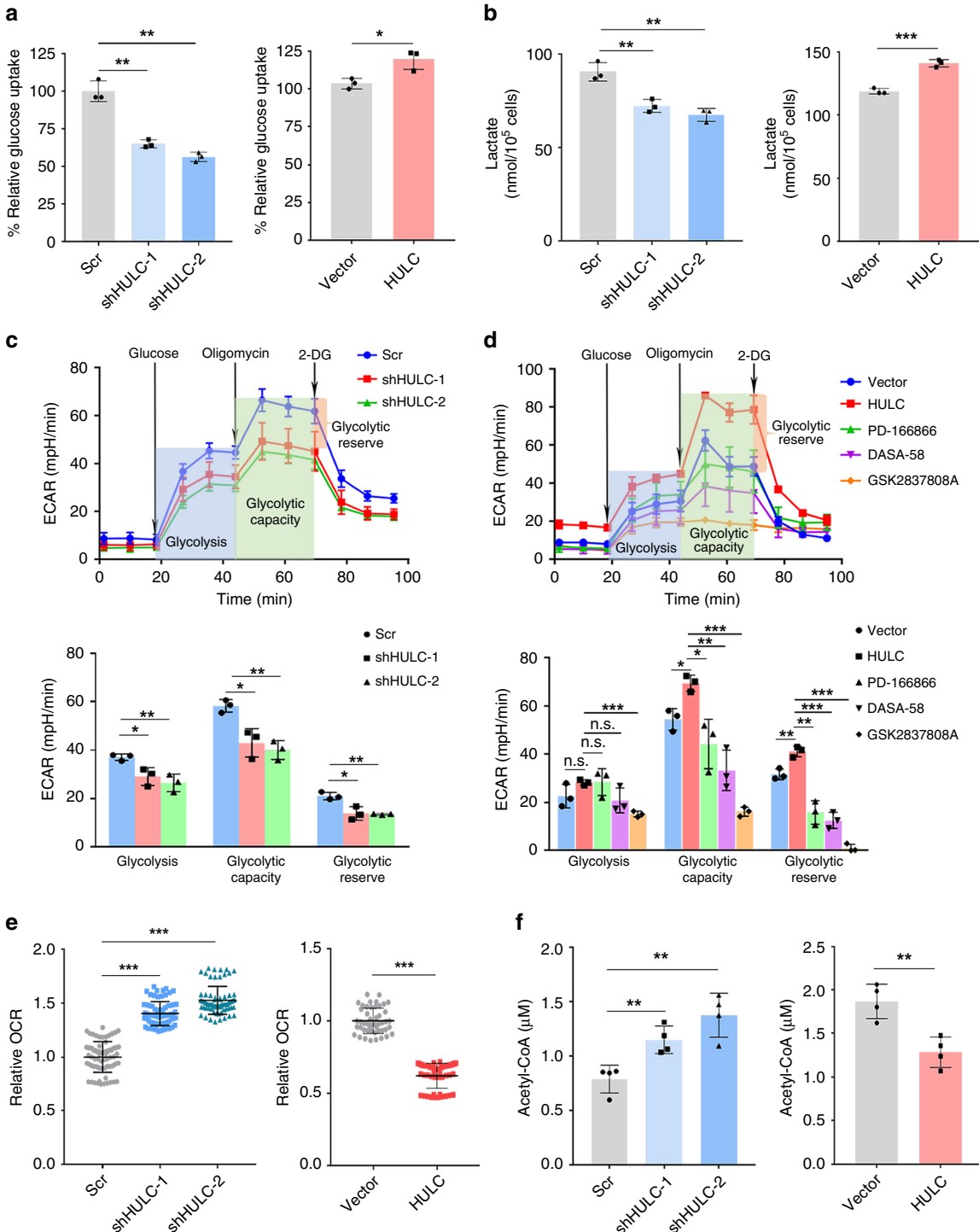

phosphorylation, and thereafter modulates their enzymatic activities.

**HULC promotes aerobic glycolysis**. Since LDHA and PKM2 both play important roles in the aerobic glycolysis process, it is only reasonable to believe that the interactions between HULC and these two glycolytic enzymes could affect the glycolysis metabolic process in the cell. To test this hypothesis, we measured glucose uptake, lactate production, and glycolytic flux in HCC cells with HULC upregulation or downregulation. We found that the knockdown of HULC reduced glucose uptake and lactate production in HepG2 cells, and the overexpression of HULC

promoted glucose uptake and lactate production (Fig. 6a, b). Similar results were observed for two other HCC cell lines, including 97L and Huh7 (Supplementary Fig. 6).

Next, we examined the glycolytic flux by measuring the extracellular acidification rate (ECAR) using the Seahorse Analyzer. As shown in Fig. 6c, d, glucose was first added to boost the glycolysis level, and the addition of ATP synthase inhibitor oligomycin shut down oxidative phosphorylation, allowing the measurement of glycolytic capacity. The following addition of glycolysis inhibitor 2-deoxy-ᴅ-glucose (2-DG) inhibited glycolysis and allowed us to evaluate the glycolytic reserve. The analysis results showed that the knockdown of

**Fig. 6 HULC promotes glycolysis. a** The relative glucose uptake measured in HepG2 cells with HULC knockdown (left panel) or overexpression (right panel). **b** Lactate production in HepG2 cells with HULC knockdown (left panel) or overexpression (right panel). Levels of lactate in the culture medium were measured and normalized to the cell number. In **a** and **b**, statistical analysis was performed by the two-sided Student's t test. Bars, mean; error bars, s. d. (n = 3 independent experiments, *P < 0.05, **P < 0.01, ***P < 0.001). **c** Glycolysis flux was examined by measuring the extracellular acidification rate (ECAR) using the Seahorse analyzer. Glucose (10 mM), ATP synthase inhibitor oligomycin (1 μM), and glycolysis inhibitor 2-DG (50 mM) were added to the cells at the indicated time points. The values of glycolysis, glycolytic capacity and glycolytic reserve were calculated by the Seahorse XF24 software. **d** The glycolysis analysis of HepG2 cells overexpressing HULC treated by FGFR1 inhibitor PD166866 (2.5 μM), PKM2 activator DASA-58 (30 μM), or LDHA inhibitor GSK2837808A (10 μM). DMSO was added instead in the control groups. In **c** and **d**, statistical analysis was performed by the two-sided Student's t test. Bars, mean; error bars, s.d. (n = 3 independent experiments, *P < 0.05, **P < 0.01, ***P < 0.001; n.s., not significant). **e** The relative oxygen consumption rates (OCR) in HepG2 cells with HULC knockdown (left panel) or overexpression (right panel). The OCR values were measured using the Seahorse analyzer. Statistical analysis was performed by the two-sided Student's t test. Data represent one of triplicate independent experiments. Bars, mean; error bars, s.d. (**P < 0.01, ***P < 0.001). **f** The relative levels of acetyl-CoA in HepG2 cells with HULC knockdown (left panel) or overexpression (right panel) as compared with the same number of corresponding control cells. Bars, mean; error bars, s.d. (n = 4 independent experiments, **P < 0.01, by the two-sided Student's t test). See also Supplementary Fig. 6. Source data are provided as a Source Data file.

HULC by two shRNAs reduced the overall glycolytic flux in HepG2 cells. Glycolysis, glycolytic capacity, and glycolytic reserve were all significantly reduced by HULC knockdown (Fig. 6c). Meanwhile, the overexpression of HULC increased the glycolysis level in HepG2 cells, and treatment with FGFR1 inhibitor, PKM2 activator or LDHA inhibitor could all abolish such effect of HULC (Fig. 6d). In summary, these data suggest that HULC positively regulates glycolysis in the cell through LDHA, PKM2, and FGFR1.

Furthermore, we investigated whether HULC could also affect the level of oxidative phosphorylation. The results showed that the levels of oxygen consumption and concentrations of acetyl-CoA were both increased by the knockdown of HULC and reduced by its overexpression (Fig. 6e, f). Taken together, the data indicate that HULC impairs oxidative phosphorylation by increasing the conversion of pyruvate to lactate instead of acetyl-CoA for TCA cycle.

**HULC enhances cell proliferation through glycolysis.** HULC has been reported to promote HCC cell proliferation[19,20]. Consistent with the literature, growth curve assay showed that cell proliferation was decreased by HULC knockdown and increased by HULC overexpression (Fig. 7a). In addition, the results from CCK8 assay showed the same tendency (Fig. 7b), and the treatment with FGFR1 inhibitor, PKM2 activator or LDHA inhibitor decreased the proliferation of HepG2-HULC cells to similar levels as the control cells (Fig. 7b). These data show that HULC promotes cell proliferation in liver cancer cells through LDHA, PKM2, and FGFR1.

To further determine whether the newly discovered function of HULC in modulating glycolysis contributes to its regulatory role in cell proliferation, we examined the effects of HULC on cell proliferation in the presence of two metabolic inhibitors, including glycolysis inhibitor 2-DG and ATP synthase inhibitor oligomycin. The difference in cell proliferation between the HepG2-shHULC cells and control cells was reduced by 2-DG treatment in a dose-dependent manner (Fig. 7c). Consistently, HepG2 cells overexpressing HULC were more sensitive to the glycolysis inhibitor 2-DG as compared with the control cells (Fig. 7d). Meanwhile, HULC reduce the sensitivity of the cells to oligomycin treatment as compared with the control cells (Fig. 7e, f). Moreover, we investigated the effects of 2-DG and oligomycin with a boarder range of concentration and calculated their half maximal inhibitory concentrations (IC50) in HepG2 cells with HULC overexpression. As a result, HepG2-HULC cells had significantly lower IC50 of 2-DG and higher IC50 of oligomycin as compared to the control cells (Fig. 7g, h).

Finally, we investigated the function of HULC in vivo. Consistent with the results from cell assays, the overexpression of HULC increased tumor growth in mice (Fig. 7i), and more lactate was detected from the tumors formed by HULC overexpressing cells than by control cells (Fig. 7j). Furthermore, we investigated the expression of HULC in clinical samples by analyzing the RNA-sequencing data from TCGA database. The analysis results show that HULC is significantly upregulated in HCC tumor samples as compared with the adjacent non-tumorous tissues (Supplementary Fig. 7a). Survival rate analysis indicates that the expression level of HULC is not significantly correlated with the survival of patients with late stage of HCC (T3 and T4). However, the early-stage (T1 and T2) HCC patients with high expression of HULC have lower overall survival probability as compared with HCC patients with low expression of HULC (Supplementary Fig. 7b), suggesting that HULC may be used as a prognosis biomarker for early-stage HCC. Taken together, the results suggest that HULC elevates glycolysis and promotes cell proliferation both in vitro and in vivo.

## Discussion

Screening for unknown lncRNA interacting proteins is crucial to understand the biological functions of lncRNA molecules. However, the available methods to study lncRNA-protein interactions all have their limitations. The TOBAP-MS method presented here provides an alternative and convenient strategy to identify novel lncRNA interacting proteins in vivo. Using this method, we identified a number of novel potential interacting proteins for lncRNA HULC. These data are important to further understand the biological functions of HULC and to unveil the underlying molecular mechanisms. To be noticed, the interacting protein data obtained via the high throughput quantitative proteomic analysis in this study require further validations.

Bioinformatic analysis shows that HULC interacting proteins are enriched in several major biological functions, e.g., response to virus and metabolism. These sub-clusters are highly connected with each other, which suggest that HULC may regulate the cross-talk among different biological processes. HULC was found to potentially interact with multiple proteins involved with the response to virus. This observation is particularly interesting because several evidences have suggested the link between HULC and hepatitis B virus (HBV) infection, a crucial factor related with HCC development. HBV-producing HCC cells express higher levels of HULC compared with their parental cell lines that do not produce HBV[40]. Moreover, HBV X protein (HBx), an oncogenic viral protein, positively correlates with the levels of HULC in HCC tumor tissues, and HBx induces the transcription of HULC via transcription factor CREB[18,19]. However, the roles of HULC in virus response and the underlying mechanisms are still unknown. This study provides new evidence for the involvement of HULC in virus response through lncRNA-protein interactions.

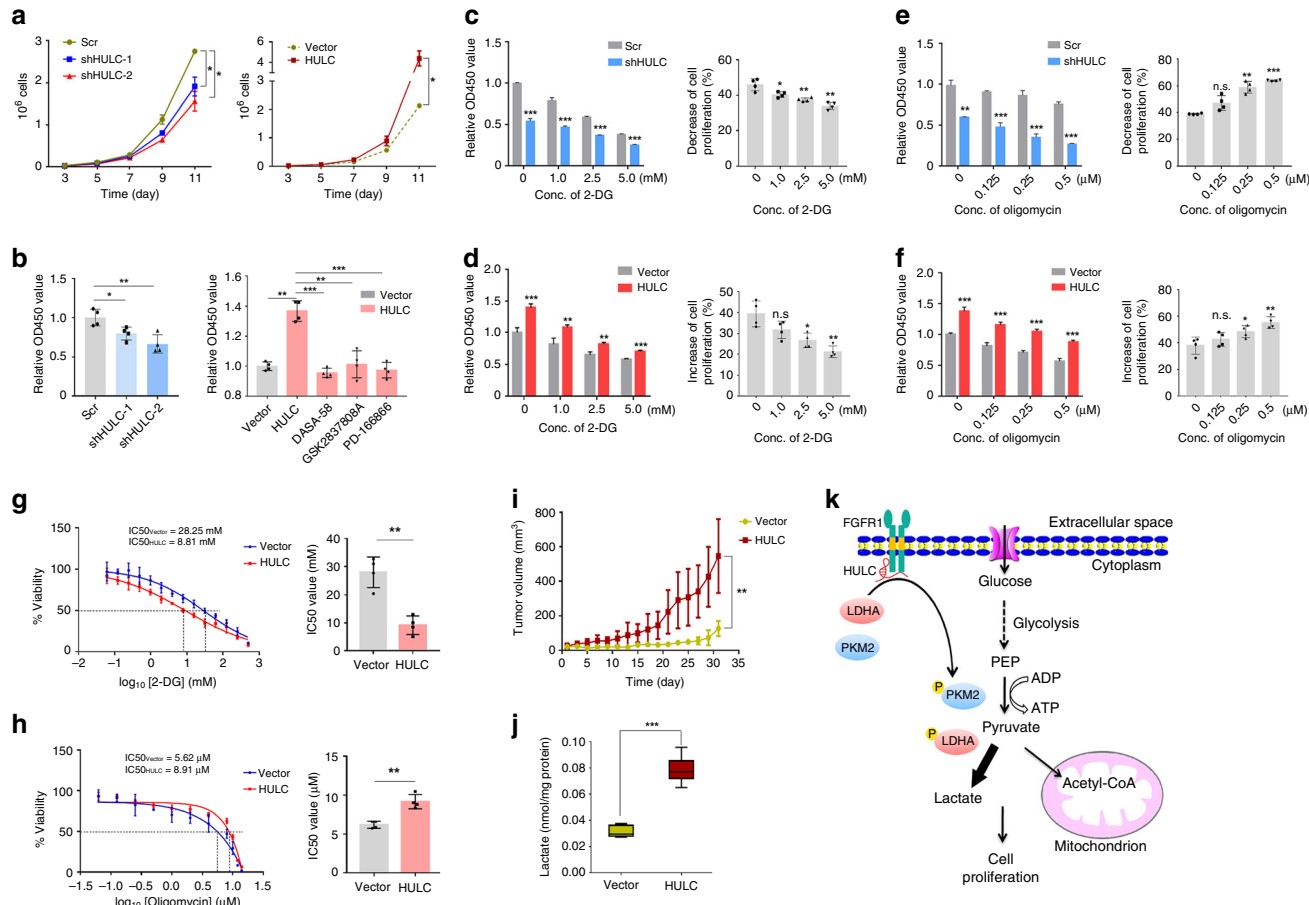

**Fig. 7 HULC promotes cell proliferation by elevating aerobic glycolysis. a** The growth curve of HepG2 cells with HULC knockdown (left panel) or overexpression (right panel). Statistical analysis was performed by one-way analysis of variance (ANOVA) (*$P < 0.05$). **b** Proliferation of HepG2 cells with HULC knockdown (left panel) or overexpression (right panel) was measured with a CCK8 assay. For rescue experiments, HepG2-HULC cells were treated by FGFR1 inhibitor PD 166866 (2.5 μM), PKM2 activator DASA-58 (30 μM), or LDHA inhibitor GSK2837808A (10 μM), respectively. DMSO was added instead in the control groups. HepG2 cells with HULC knockdown (**c**) or overexpression (**d**) were incubated with indicated concentrations of 2-DG, and cell proliferation was measured with the CCK8 assay. HepG2 cells with HULC knockdown (**e**) or overexpression (**f**) were incubated with indicated concentrations of oligomycin, and cell proliferation was measured with the CCK8 assay. IC50 values of 2-DG (**g**) and oligomycin (**h**) in HepG2-HULC cells were calculated by dose curve analysis. In **b**–**h**, statistical analysis was performed by two-sided Student's $t$ test. Bars, mean; error bars, s.d. ($n = 4$ independent experiments, *$P < 0.05$, **$P < 0.01$, ***$P < 0.001$, n.s., not significant). **i** HepG2 cells expressing empty vector or HULC were injected subcutaneously into nude mice, and tumor volumes were measured every 2 days ($n = 4$ for each group). Bars, mean; error bars, s.d. (***$P < 0.001$, by one-way ANOVA). **j** The levels of lactate in the tumors were measured and normalized using protein concentrations ($n = 13$ tissues). The box plot includes data between the 25th and 75th percentiles, with the horizontal line representing the median. The upper whisker is the maxima, and the lower whisker is the minima. Bars, mean; error bars, s.d. (***$P < 0.001$, by two-sided Student's $t$ test). **k** The proposed molecular mechanism by which HULC regulates cell proliferation through modulating glycolysis. See also Supplementary Fig. 7. Source data are provided as a Source data file.

Further investigation is required to confirm such observation and to clarify the possible molecular mechanisms.

Altered cellular metabolism is a major hallmark of cancer, and elevated glycolysis exemplified by high glucose consumption and lactate production has been observed in many types of tumor cells, which supports biosynthesis to sustain tumor growth[1,5]. Recent studies have implicated the roles of lncRNAs in the regulation of glycolysis. Some lncRNAs are reported to modulate the expression levels of glycolytic enzymes[41]. For example, lncRNA-PVT1 promotes glycolysis by elevating the expression of hexokinase-2 (HK2) through acting as molecular sponge to repress miR-497, a miRNA that targets HK2[42]. Some lncRNAs are known to regulate glycolysis related transcription factors or signaling pathways. For example, lincRNA-p21 has been reported to promote glycolysis under hypoxic conditions through HIF-1α[15]. In addition, lncRNA-NRCP enhances glycolysis through the STAT1 pathway in ovarian cancer[43]. Moreover, the prostate

cancer-specific lncRNA prostate cancer gene expression marker 1 alters tumor metabolism by coactivating the androgen receptor and transcriptional factor c-Myc[44]. These studies highlight the evolving roles of lncRNAs in glycolysis, but majority of these lncRNAs play their roles through regulating gene expression. However, our findings reveal a unique mechanism that lncRNA-HULC regulates glycolysis by direct binding to glycolytic enzymes and modulating their enzymatic activities without affecting their expressions, which expands our understanding of the mechanisms by which lncRNA molecules exert their regulatory functions, especially in cellular metabolism.

LDHA promotes the conversion of pyruvate to lactate, and its activity positively correlates with the Warburg effect[6,35]. The enzymatic activity of LDHA is modulated by its posttranslational modifications, including acetylation and phosphorylation among many others[35,45]. Fan et al. reported that oncogenic receptor tyrosine kinase FGFR1 directly phosphorylates LDHA, which

increases the binding of LDHA to its substrate NADH and enhances its enzymatic activity[35]. Here, we observed increased phosphorylation of LDHA and enhanced LDH activity induced by the overexpression of HULC.

PKM2 is also a key regulator of aerobic glycolysis. PKM2 is highly expressed in human cancer, and replacing PKM2 with PKM1 in the constitutively active tetrameric form leads to a reversal of the Warburg effect and reduces tumorigenesis[8]. The low activity of PKM2 promotes the Warburg effect by favoring the accumulation of metabolic intermediates that can be used for biosynthesis, and therefore promotes cell growth and proliferation[46,47]. Cells have multiple ways of regulating the activity of PKM2. For example, binding with small molecules, e.g., fructose 1, 6-bisphosphate (FBP), serine, and phosphatidylserine, could activate PKM2, whereas binding with certain amino acids, such as alanine and phenylalanine, inhibits its enzymatic activity[48]. Posttranslational modifications, i.e., acetylation and methylation, could also regulate PKM2[49,50]. Phosphorylation of PKM2 on Y105 by the intracellular kinase domain of FGFR1 prevents the binding of FBP to PKM2 and inhibits the formation of active tetramer[38]. In this study, increased Y105 phosphorylation and its less tetrameric form was observed for PKM2 in HULC overexpressing cells, which was accompanied by reduced PK activity.

Based on the results, we propose a molecular mechanism model that depicts the role of HULC in the regulation of glycolysis (Fig. 7k). HULC directly binds with LDHA and PKM2, enhances their interactions with the intracellular domain of the upstream kinase FGFR1, elevates their phosphorylation, and modulates their enzymatic activities. The elevation of LDHA activity and decrease of PKM2 activity both contribute to higher level of glycolysis and thereby promote cell proliferation. This study reveals a unique mechanism by which the lncRNA molecule directly impacts cellular mechanism by regulating the localizations and activities of metabolic enzymes.

## Methods
**Antibodies and reagents**. Mouse monoclonal antibody against β-actin (Cat# T0022; 1:3000) was from Affinity Biosiences (Cincinnati, OH, USA). Rabbit polyclonal antibody against PKM1 (Cat# 15821-1-AP; 1:1000), PKM2 (Cat# 15822-1-AP; 1:30 for IP), LDHA (Cat# 19987-1-AP; 1:30 for IP), LDHB (Cat# 14824-1-AP; 1:2000), TGM2 (Cat# 15100-1-AP; 1:1000), DDX58 (Cat# 20566-1-AP; 1:300), LGALS3BP (Cat# 10281-1-AP; 1:500) and mouse monoclonal antibody against FGFR1 (Cat# 60325-1-Ig; 1:1000) were from Proteintech (Chicago, IL, USA). Rabbit monoclonal antibodies against PKM2 (Cat# 4053S; 1:1000) and GAPDH (Cat# 5174T; 1:1000); rabbit polyclonal antibodies against mTOR (Cat# 2972S; 1:1000), p-LDHA (Y10) (Cat# 8176S; 1:1000) and p-PKM2 (Y105) (Cat# 3827S; 1:1000) were from Cell Signaling Technology (Danvers, MA, USA). Mouse monoclonal antibodies against LDHA (Cat# sc-137243, 1:100 for IF) and PKM2 (Cat# sc-365684, 1:100 for IF) were bought from Santa Cruz Biotechnology (Santa Cruz, CA, USA). Mouse monoclonal antibody against LDHA (Cat# ab85326, 1:50 for IP); rabbit monoclonal antibodies against LDHA (Cat# ab101562; 1:1000), FGFR1 (Cat# ab76464, 1:190 for IP), sodium potassium ATPase (Cat# ab76020; 1:5000), Flag (Cat# 205606; 1:1000) and PDIA3 (also called ERp57, Cat# ab154191; 1:1000); polyclonal antibody against histone H2A (Cat# ab18255; 1:1000); recombinant human PKM2 (rPKM2), LDHA (rLDHA), and FGFR1 intracellular domain (rFGFR1 AA399-822) were bought from Abcam (Cambridge, MA, USA). Alexa Fluor 488 AffiniPure goat-anti-mouse IgG (H + L) (Cat# A-11034, 1:100 for IF) and Alexa Fluor 546 AffiniPure goat-anti-rabbit IgG (H + L) (Cat# A-10040, 1:100 for IF) were from Thermofisher Scientific (Waltham, MA, USA). Recombinant human FGF was from Peprotech (London, UK). FGFR1 kinase inhibitor PD166866, PKM2 activator DASA-58, and LDHA inhibitor GSK2837808A were from MedChemExpress (Monmouth Junction, NJ, USA). Dithiothreitol (DTT), iodoacetamide (IAA), urea, 2-deoxyglucose (2-DG), 2-(N-(7-nitrobenz-2-oxa-1,3-diazol-4-yl) -amino)−2- deoxyglucose (2-NBDG), and tobramycin were from Sigma (St. Louis, MO, USA). Oligomycin was bought from Toronto Research Chemicals (Toronto, Ontario, Canada). Dynabeads® His-tag isolation kit, Lipofectamine 2000, BCA reagents, Protein A and G magnetic beads were purchased from Invitrogen (Grand Island, NY, USA). Enhanced chemiluminescence reagents were purchased from Pierce Biotechnology (Rockford, IL, USA). Protease Inhibitor Cocktail tablets were purchased from Roche Diagnostics (Indianapolis, IN, USA). RNasin® ribonuclease inhibitors and sequencing grade modified trypsin were

purchased from Promega (Madison, WI, USA). LC-MS grade acetonitrile was from Merck (White-house Station, NJ, USA). Water used in this study was deionized using a Milli-Q purification system (Millipore, Billerica, MA, USA).

**Cell Lines**. HCC cell lines, including HepG2, MHCC97L, and Huh7 cells, were maintained in our laboratory with Dulbecco's modified Eagle's medium (DMEM) supplemented with 10% fetal bovine serum and 1% penicillin–streptomycin (100 μg mL$^{-1}$) at 37 °C in a humidified atmosphere with 5% $CO_2$.

**SILAC Labeling**. HepG2 cells were cultured using the SILAC DMEM (Thermo-Fisher Scientific) supplemented with 10% dialyzed fetal bovine serum and 1% penicillin–streptomycin. The cells transfected with HULC-J6f1 plasmids were cultured in normal lysine and arginine containing medium, and the cells transfected with J6f1 vector plasmids were cultured in a medium containing both [$^{13}$C6]-L-lysine and [$^{13}$C6]-L-arginine. The cells were cultured for more than seven generations, and the same amount of proteins from light labeled and heavy labeled cells were mixed and analyzed by mass spectrometry to evaluate the labeling efficiency before use.

**Plasmids and cloning**. The total RNA was isolated from cells using Trizol reagent (Invitrogen), and complementary DNA (cDNA) was synthesized using an FastQuant RT kit (TianGen, Beijing, China) according to the manufacturer's instructions. The cDNA of exon 10 and exon 9 of the *PKM* gene was amplified via a polymerase chain reaction (PCR) from creating flanking EcoRI and XhoI restriction sites using Premix Taq DNA Polymerase (TaKaRa, Otsu, Japan) and was cloned into the pcDNA3.1(+) vector (Invitrogen). For the overexpression of HULC-J6f1 and HULC, the cloning vectors of *HULC-J6f1* and *HULC* were constructed by GENEWIZ (Suzhou, China) and then were subcloned to the pCDH expression vector (System Biosciences, Palo Alto, CA, USA) upon double restriction enzyme digestion by EcoRI and XbaI. The plasmids of full-length wild-type *LDHA* and *PKM2* were constructed using a pCDNA3.1-CMV vector with a C-terminal 3xFlag tag (SyngenTech, Beijing, China). For the construction of LDHA-Y83F and Y10F mutants, overlapped fragments were amplified by PCR with primers containing mutations and inserted into pCDNA3.1-CMV vector by a homolog recombination kit (YiSheng, Shanghai, China) between the NdeI (NEB) and BamHI (NEB) sites. The *PKM* mutations, including Y83F, Y105F, and Y370F, were generated by overlap extension PCR and subcloned between the NdeI (NEB) and EcoRV (NEB) sites of the pCDNA3.1-CMV vector. All constructs were verified by DNA sequencing. Sequences of primers in plasmid construction were shown in Supplementary Table 3.

For knockdown experiment, recombinant genes coding shRNA against target genes and the nontargeting control shRNA (Supplementary Table 4) were constructed to the pLKO.1-TRC cloning vector (Addgene, Cambridge, MA, USA). The production of lentivirus and cell infection was carried out following the manufacture's protocol (www.addgene.org/plko). The knockdown efficiency was confirmed by qRT-PCR or western blotting.

**RNA Isolation and qRT-PCR**. Quantitative mRNA analysis was performed on a 7500 Fast Real-Time PCR System (ABI, Foster City, CA, USA) using the SuperReal SYBR Green PreMix (TianGen, Beijing, China) following the manufacturer's instructions. The mean Ct for each sample was normalized using 18s rRNA as the reference gene (for primer sequences, see Supplementary Table 5).

**In vitro transcription**. The templates for in vitro transcription were obtained by PCR, and the primers of HULC containing the T7 promoter sequence (TAATACGACTCACTATAGGG) were purchased from Invitrogen (for primer sequences, see Supplementary Table 6). The PCR products were examined with 2% agarose gel electrophoresis. The target bands were cut out and purified with an agarose Gel DNA Extraction Kit Ver.4.0 (TaKaRa, Otsu, Japan). Biotin-labeled sense and antisense chains of *HULC* were transcribed in vitro using biotin-16-UTP (Epicentre, Madison, WI) with a MEGAscript Kit (Life Technologies, Carlsbad, CA). The synthesized RNAs were purified using a MEGAclear Kit (Invitrogen) following the manufacturers' instructions. The integrity and size of the synthesized RNAs were evaluated using agarose gel electrophoresis, and the incorporation of biotin was detected by biotin-HRP dot blot with a chemiluminescent biotin-labeled nucleic acid detection kit (Beyotime, Shanghai, China) following the manufacturer's instructions.

**RNA FISH and Immunofluorescence**. Cells were plated in 12-well plates containing sterile glass coverslips, allowed to grow overnight. Then the cells were fixed with 37% formaldehyde in PBS for 10 min at RT, followed by permeabilize in 70% ethanol for 1 h at 4 °C. Slides were hybridized at 37 °C for 14–16 h. The antisense probes were dissolved at 20 nM in hybridization buffer for RNA FISH. After overnight hybridization, slides were washed in 10% formamide/2× SSC at 37 °C for 30 min in a shaker followed by staining with Alexa Fluor 546-conjugated streptavidin for 1 h at 37 °C. The slides were washed in 10% formamide/2× SSC at 37 °C for 30 min in a shaker and then washed in PBS with 0.1% (v/v) Tween 20 three times and then processed for Immunofluorescence. The cells were blocked in 3%

BSA in PBS for 1 h at RT and then incubated with primary antibodies at 4 °C overnight, followed by staining with Alexa Fluor 488-conjugated secondary antibodies for 1 h at RT. The coverslips were counterstained with DAPI, mounted with ProLong Gold antifade reagents and visualized with confocal laser scanning microscopy (Olympus, Tokyo, Japan).

**Tobramycin affinity purification**. N-hydroxysuccinimide-activated Sepharose 4 Fast Flow beads (GE Healthcare Life Science, Marlborough, MA, USA) were derivatized with 40 mM tobramycin following the manufacturer's protocol. For affinity purification, the $4\times$ binding buffer (80 mM Tris, pH 7.4/4 mM $CaCl_2$/4 mM $MgCl_2$/2 mM DTT) was freshly prepared. The tobramycin-coated sepharose beads were blocked with 500 μL of blocking buffer ($1\times$BP/300 mM KCl/0.1 mg mL$^{-1}$ tRNA/ 0.5 mg mL$^{-1}$ BSA/ 0.01% Triton X-100) at 4 °C overnight. The beads were collected and incubated with cell lysates from $2 \times 10^7$ cells at 4 °C for 4 h. The beads were then isolated by centrifugation and washed with 1 mL of washing buffer (40 mM Tris/120 mM NaCl/1% TritonX-100, pH = 7.4) at room temperature for ten times. The bound RNA and proteins were eluted with 30 mM tobramycin solution. The HepG2-J6f1 cells were processed in parallel as the negative control. The proteins purified from HepG2-HULC-J6f1 cells were mixed with the control sample, separated on a 10% SDS-PAGE, and visualized using silver staining. Each gel lane was diced into ten slices, and in-gel tryptic digestion was conducted. Three biological replicates were analyzed for quantification.

**LC–MS/MS and data analysis**. The in-gel digested samples were desalted using C18 ZipTip and loaded on a nanoUPLC system (Waters, Milford, MA, USA) equipped with a self-packed C18 column (C18, $150 \times 0.075$ mm, 1.7 μm). The peptides were eluted using a 5–40% B gradient (0.1% formic acid in acetonitrile) over 90 min into a nanoelectrospray ionization LTQ Fusion mass spectrometer (ThermoFisher Scientific). The Xcalibur™ software Version 4.1 (ThermoFisher Scientific) was used to set the instrumental parameters and collect the data. The mass spectrometer was operated in data-dependent mode, in which an initial FT scan recorded the mass range of $m/z$ 350–1500. The spray voltage was set between 1.8 and 2.0 kV, and the mass resolution used for the MS scan was 60,000. The top 20 most intense masses were selected for collision-induced dissociation (CID) with normalized collision energy of 35. Only those precursors with charge state +2 or higher were sampled for MS$^2$. The dynamic exclusion duration was set to 45 s with a 10-ppm tolerance around the selected precursor. The AGC target value and maximum injection time were set as 1E6 and 100 ms for MS scans and 5E3 and 125 ms for MS$^2$ scans.

Raw data were searched against the Swiss-Prot/UniProt human protein database (release August 2018) containing 20,325 sequence entries and a common contaminants database using the Andromeda search engine embedded in the MaxQuant Software (version 1.6.3.4). The following parameters were applied during the database search: 20 ppm precursor and 0.5 Da fragment mass error tolerance, Arg/Lys (+6.0201 Da, SILAC heavy amino acid) as variable modifications, static modifications of carbamidomethylation for all cysteine residues, flexible modification of oxidation modifications for methionine residues, and two missed cleavage site of trypsin was allowed. The false discovery rate < 0.01 was used as filtering criteria for all identified proteins.

Protein quantitation was performed in R (3.5.2) by using the unique peptide intensities exported from the Maxquant. Protein identifications classified as Only Identified by site, contaminants and reverse were excluded from the data frame. In addition, proteins identified with only one unique peptide were also discarded. Student's $t$-test was performed for statistical analysis, and proteins with log$_2$ ratio $_{HULC/control} \geq 1.5$ and $P$ value $\leq 0.05$ were considered as potential HULC interacting proteins. Volcano plot was plotted with the R package ggplot2 (3.1.0). KEGG pathway enrichment analysis was acquired through the R package clusterProfiler (v3.10.1). Known PPIs were obtained from String 11 (https://string-db.org/) and integrated in Cytoscape 3.7 for visualization.

**Western blotting**. The cellular proteins were resolved by SDS-PAGE and transferred to the Immobilon-P membrane (0.2 μm pore size, Millipore, Billerica, MA, USA). Primary antibodies were incubated with the membranes at 4 °C overnight (BD Biosciences, San Jose, CA, USA). Then, the blots were incubated with the horseradish peroxidase-conjugated secondary antibody and developed by enhanced chemiluminescence. β–actin was used as the internal standard. The band intensities were measured using ImageJ for quantitative comparisons.

**RNA pull-down**. Cellular proteins were extracted from HepG2 cells with lysis buffer (40 mMTris, 120 mM NaCl, 1% Triton X-100, 1 mM NaF, and 1 mM $Na_3VO_4$) supplemented with $1\times$ protease inhibitor cocktail (Roche Diagnostics, Mannheim, Germany) and 1 U mL$^{-1}$ RNase inhibitor. The total protein concentration of the extract was measured with a BCA assay. Then, 40 μL of MyOne™ Streptavidin C1 Dynabeads (Invitrogen) were washed and incubated with 6 μg of biotinylated sense or antisense HULC RNA in a binding buffer (50 mM $Na_3PO_4$, 300 mM NaCl, 0.01% Tween-20, pH = 8.0) at 4 °C for 3 h. Next, the beads were added to cell lysates and incubated at 4 °C for 4 h. Then, the beads were washed five times with lysis buffer, mixed with an SDS-PAGE sample loading buffer, and analyzed by western blotting.

**RNA immunoprecipitation (RIP)**. Cellular proteins from HepG2 cells were extracted with lysis buffer (40 mM Tris, 120 mM NaCl, 1% Triton X-100, 1 mM NaF, 1 mM $Na_3VO_4$) supplemented with $1\times$ protease inhibitor cocktail and the RNase inhibitor. Then, the total protein concentration of the extract was measured with BCA assay. The proteins of interest along with the binding RNAs were isolated with the corresponding primary antibodies using protein A/G magnetic beads. The same amount of cellular proteins was assayed simultaneously for IgG control. Then, the co-precipitated RNAs were extracted with Trizol, and the amount of HULC in the eluate was analyzed by qRT-PCR. LincRNA-p21 was analyzed as an irrelevant RNA control.

**His-tag pull-down assay**. The His-tagged rPKM2 or rLDHA was incubated with Dynabeads® His-tag isolation magnetic beads (Invitrogen) in binding buffer (50 mM $Na_3PO_4$, 300 mM NaCl, 0.01% Tween-20, pH = 8.0) at 4 °C for 4 h. Unbound protein was removed, and the protein-coupled beads were incubated with in vitro transcribed HULC RNA for 2 h in a pull-down buffer (3.25 mM $Na_3PO_4$, 70 mM NaCl, 0.01% Tween-20, pH = 7.4) at 4 °C; next, the beads were washed five times with binding buffer supplemented with a protease inhibitor cocktail and RNase inhibitor. Finally, the copurified HULC was extracted with the Trizol reagent and was analyzed by qRT-PCR.

**Molecular interaction analysis**. SPR binding assay was performed by using a Biacore™ T200 system. The purified biotinylated HULC was immobilized on a streptavidin Biacore chip™ at 4 μg mL$^{-1}$ in PBS. rLDHA was diluted to different concentrations with PBS, and the flow rate was maintained at 30 μL min$^{-1}$ throughout the kinetics experiment. The contact time was settled for 120 s and dissociation time was kept at 200 s. The collected data were analyzed using the Biacore T200 Evaluation Software 2.0 (GE Healthcare) and GraphPad Prism 7. Bio-Layer Interferometry analysis was performed by using the Octet RED96 system (ForteBio, Fremont, CA). The purified biotinylated HULC was immobilized on a Streptavidin Biosensor (ForteBio) in 16 μg mL$^{-1}$ in PBS with 0.02% Tween 20 and 0.1% BSA. rPKM2 protein was diluted to different concentrations with PBS with 0.02% Tween 20 and 0.1% BSA. The association process was performed for 120 s, and the dissociation process was performed for 200 s. The resulting data were analyzed after subtracting background and the equilibrium dissociation constant $K_d$ was calculated using Octet RED96 analysis software 7.0 and GraphPad Prism 7.

**Membrane protein extraction**. HepG2 cells were infected with vector or HULC-containing plasmid and cultured for 48 h. The membrane proteins were isolated using the Minute Plasma Membrane Protein Isolation Kit (Invent Biotechnologies, Eden Prairie, MN, USA) according to the manufacturer's instructions. Briefly, $5 \times 10^7$ cells were homogenized in the supplied buffer A supplemented with the protease inhibitor cocktail and the RNA ribonuclease inhibitor. The collected cytosol and plasma membrane fractions were used for western blotting. GAPDH and sodium potassium ATPase were used as cytoplasm and plasma membrane-loading controls, respectively.

**Enzyme activity measurements**. PK activity was examined using a pyruvate kinase activity assay kit (BioVision, Mountain View, CA, USA) according to the manufacturer's protocol. Briefly, $10^6$ cells were homogenized in 0.5 mL cold assay buffer, and the supernatant was collected by centrifugation. The assay was carried out with 50 μL of diluted cell lysate mixed with 44 μL of assay buffer, 2 μL substrate mix, 2 μL of enzyme mix and 2 μL of detection probe solution. The increase in the absorbance at 570 nm between $T_{15 min}$ and $T_{0 min}$ was measured by a microreader to calculate the PK activity in the sample. For the assessment of LDH activity, the cells were lysed in a similar manner, and 50 μL of diluted cell lysate was examined by a LDH activity assay kit (BioVision) according to the manufacturer's protocol. The increase in the absorbance at 450 nm between $T_{30 min}$ and $T_{0 min}$ was measured by a microreader to calculate the LDH activity in the sample.

**Measurements of glucose uptake and metabolites**. The fluorescent 2-DG analog 2-NBDG was used to measure glucose uptake. The cells were incubated with 2-NBDG (10 μM) for 1 h, washed twice by PBS, and analyzed by flow cytometry (BD Biosciences, Franklin Lakes, NJ). For assessment of lactate production, the cells were resuspended in DMEM without pyruvate, plated in 96-well plates ($8 \times 10^4$ cells per well), and allowed to grow for 6 h. The cell media was collected and diluted 1:6 in lactate assay buffer. The amount of lactate in the media was then measured using the lactate colorimetric assay kit (BioVision) according to the manufacturer's instructions. The amount of lactate in the sample was calculated by subtracting the amount of lactate in the media. For the measurement of acetyl-CoA, $2 \times 10^6$ cells were lysed with 500 μL assay buffer and centrifuged for 15 min at 10,000 g. Then, the supernatants were deproteinized by adding 2 μL of 1 N perchloric acid/mg protein, and the resulting supernatant was neutralized by adding 3 M KHCO$_3$. The amount of acetyl-CoA was examined using a PicoProbe acetyl-CoA assay kit (BioVision), according to the manufacturer's instructions.

**Oxygen consumption and ECAR**. The oxygen consumption rate (OCR) and ECAR of HepG2 cells were determined using the Seahorse XF extracellular flux

analyzer (Agilent, Santa Clara, CA, USA). The cells were plated at a density of $3\times10^4$ cells per well on 24-well Seahorse plates and were allowed to attach overnight in growth medium. Then, the adherent cells were washed, and a fresh assay medium was added before analysis. The cartridge was loaded to dispense glucose and metabolic inhibitors sequentially at specific time points: glucose (10 mM), followed by oligomycin (1 μM), and then 2-DG (50 mM). OCR and ECAR were measured and plotted by the Seahorse XF24 software.

**Cell proliferation assays.** For the growth curve assay, the cells were plated in a six-well plate with a density of $2 \times 10^5$ cells per well, counted manually and replated every other day. The growth curve was plotted based on the fold increase in the cell number. To measure the cell proliferation affected by metabolic inhibitors, the cells were plated at 2000 cells per well in 96-well plates and incubated with oligomycin or 2-DG with different concentrations for 48 h at 37 ℃. The cells were measured with the CCK8 assay (Dojindo, Kumamoto, Japan) according to the manufacturer's protocol. Briefly, 10 μL of CCK8 solution was added to each well, and the samples were incubated at 37 ℃ for 2 h before the absorbance was measured at 450 nm.

**In vivo tumor growth assay.** In vivo proliferation assays were performed using 5-week-old female BALB/c-nude mice (Chinese Academy of Sciences, Beijing, China). Briefly, $5 \times 10^6$ cells were implanted into the subcutaneous tissues. Tumor volumes were recorded every 2 days and were calculated with the formula $V =$ (length × width$^2$)/2. After 4 weeks, the mice were executed, and the tumors were excised. The experimental protocols were evaluated and approved by the Institutional Review Board of Tianjin Medical University.

**Statistics and reproducibility.** The RNA FISH, immunofluorescence staining, agarose gel electrophoresis assay, and western blotting experiments were carried out at least three times, and representative images are shown. SPSS version 17.0 software was used for statistical analyses, and Prism version 5.0 (GraphPad) was used to generate plots. For the comparisons, two-sided Student's $t$-test was performed between two groups, and the growth curve and in vivo tumor growth data were analyzed by repeated measures analysis of variance (ANOVA). $P$ values < 0.05 were considered statistically significant.

**Reporting summary.** Further information on research design is available in the Nature Research Reporting Summary linked to this article.

## Data availability

All data generated or analyzed during this study are included in this published article and its supplementary information files, and are available from the corresponding authors upon reasonable request. The mass spectrometry proteomic data have been deposited to the ProteomeXchange Consortium via the PRIDE partner repository[51] with the dataset identifier PXD012927. The human protein sequences used for protein identification were downloaded from the UniProt database. RNA-sequencing data were downloaded from The Cancer Genome Atlas (TCGA) database by R package TCGAbiolinks (version,2.10.5). The source data underlying Figs. 1b-d, 2d, 3a, c-h, 4a, c-i, 5a-e, 6a-f and 7a-j and Supplementary Figs 3a,b, 4, 5a-l and 6a-f are provided as a Source Data file.

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

## Acknowledgements

This work was supported by grants from National Natural Science Foundation of China (21974094 and 21575103 to R.C.; 81988101, 31671421 and 81472683 to N.Z.; 81872377 to Y.L.), National Science and Technology Major Project of China (2018ZX10723204 to N.Z.), National Key Scientific Instrument and Equipment Development Project (2013YQ16055106 to N.Z.), Tianjin Natural Science Foundation (18JCYBJC25200 to R.C and 18JCYBJC25600 to Y.L.), Young Elite Scientists Sponsorship Program by Tianjin (TJSQNTJ-2017-10 to R.C.), Postgraduate Innovation Fund of 13th Five-Year Comprehensive Investment from Tianjin Medical University (YJSCX201811 to C.W.).

## Author contributions

R.C. and N.Z. conceived and designed the project. C.W., S.Y., and Y.L. carried out the molecular and cellular biology experiments with assistance from H. W., M. X., B. Y., and R. K. Mass spectrometric analyses were performed by R.C. with assistance from H.W., X.S. and G.Q. The manuscript was written by R.C. and N.Z. with assistance from C.W. and Y.L.

## Competing interests

The authors declare no competing interests.
