## [Peer Review File · Nature Communications]

Reviewers' comments:

Reviewer #1:

In the original revised manuscript of Chen, et al. [REDACTED], the authors performed RAP-MS, aiming to validate the initial findings of HULC-interacting proteins by TOBAP-MS. It is surprising that the initial TOBAP-MS identified 140 potential HULC-interacting proteins. However, RAP-MS revealed 25 proteins, and only 4 of them overlapped with the initial 140 protein targets revealed by TOBAP-MS. Particularly, RAP-MS did not identify PKM2, which was a key protein target identified by TOBAP-MS and was functionally investigated in this manuscript. It is understandable that there are certain variations between experiments and different technologies. However, the interaction between HULC and PKM2 served as the fundamental basis for the entire manuscript and cannot be observed using independent methodology, which raised the reviewer's concern.

In the current revised manuscript, the authors did not provide any additional validation of the findings illustrated in Figures 1 and 2. While the authors provided reasoning for the RAP-MS method's inability to identify PKM2 in the revised manuscript, they did not sufficiently explain the differences between the TOBAP-MS and RAP-MS methodologies and the reason for the discrepancies between the results (particularly, the differences in the identified protein targets and HULC-protein interactions). Further validating protein-ncRNA interactions would be needed to confirm the conclusion of the current manuscript. Taken together:

1. The authors should experimentally validate the 4 common proteins revealed by TOBAP-MS and RAP-MS.
2. The authors should explain the differences between the TOBAP-MS and RAP-MS methodologies to more adequately clarify the differences in the technologies.
3. The authors should either move Figures 1 and 2 to the Supplementary Dataset or remove them entirely if they are unable to provide further validation of the experimental results.
4. If they wish to validate the aforementioned data, the authors should perform a CLIP assay using antibodies targeting both PKM2 and LDHA using HULC and HULC KO.

The authors must address these concerns to ensure scientific rigor sufficient for publication of their manuscript.

Reviewer #4:

I have no additional concerns.

Reviewer #1:

In the original revised manuscript of Chen, et al. [REDACTED], the authors performed RAP-MS, aiming to validate the initial findings of HULC-interacting proteins by TOBAP-MS. It is surprising that the initial TOBAP-MS identified 140 potential HULC-interacting proteins. However, RAP-MS revealed 25 proteins, and only 4 of them overlapped with the initial 140 protein targets revealed by TOBAP-MS. Particularly, RAP-MS did not identify PKM2, which was a key protein target identified by TOBAP-MS and was functionally investigated in this manuscript. It is understandable that there are certain variations between experiments and different technologies. However, the interaction between HULC and PKM2 served as the fundamental basis for the entire manuscript and cannot be observed using independent methodology, which raised the reviewer's concern.

In the current revised manuscript, the authors did not provide any additional validation of the findings illustrated in Figures 1 and 2. While the authors provided reasoning for the RAP-MS method's inability to identify PKM2 in the revised manuscript, they did not sufficiently explain the differences between the TOBAP-MS and RAP-MS methodologies and the reason for the discrepancies between the results (particularly, the differences in the identified protein targets and HULC-protein interactions). Further validating protein-ncRNA interactions would be needed to confirm the conclusion of the current manuscript. Taken together:

Our data have shown that TOBAP-MS is more sensitive compared to RAP-MS. The discrepancies between these two methods may be caused by several reasons, such as the low endogenous level of the target lncRNA, inaccessibility of target sequence, low efficiency of UV crosslinking, etc. In addition, TOBAP isolates both direct and indirect interacting proteins, while RAP only purifies only direct binding proteins. We now add this part to the main text and the differences between these two methods are discussed. Although PKM2 is missed in RAP-MS data, we have confirmed its interaction with HULC in 8 independent assays both *in vivo* and *in vitro*, which actually proves the advantage of TOBAP. To further evaluate the method, we also test some more identified HULC-binding proteins as suggested.

1. The authors should experimentally validate the 4 common proteins revealed by TOBAP-MS and RAP-MS.

We agree that it's important to further validate the identification results. Two of these four common proteins observed by both TOBAP-MS and RAP-MS have been validated in the original manuscript, including LDHA and TGM2. As suggested by the reviewer, we further validate the other two by RNA-IP, i. e. H2A and PDIA3, and the results are shown in the revised Fig. 2c. Collectively, we have validated 7 proteins with different positions spreading in the positive region of the volcano plot (Fig. 2a), and the results prove the ability of TOBAP-MS for characterizing lncRNA binding proteins.

2. The authors should explain the differences between the TOBAP-MS and RAP-MS methodologies to more adequately clarify the differences in the technologies.

We now add description of the RAP-MS data in the main text, and explain the differences between those two methods as suggested on pg8.

3. The authors should either move Figures 1 and 2 to the Supplementary Dataset or remove them entirely if they are unable to provide further validation of the experimental results.

Figs. 1 and 2 show how we performed the experiments and what proteins were identified, and we believe that those data are important for the audience to see how we came up with the hypothesis and to better understand the results. Besides, the last part of Fig. 2 is to validate the identification results. Removing these figures entirely may greatly affect the flow of the whole manuscript. Therefore, we decide to keep some of the original figures, and modify Fig. 2 by moving the pathway analysis results to supplementary information and adding more validation results.

4. If they wish to validate the afore mentioned data, the authors should perform a CLIP assay using antibodies targeting both PKM2 and LDHA using HULC and HULC KO.

We agree that CLIP-seq is a useful technique for large-scale identification of protein binding RNAs. However, the main subject of this paper is to understand how HULC binds with these enzymes. We have demonstrated that antibodies targeting PKM2 or LDHA could bind with HULC by RNA-IP, which serves similar purpose as CLIP-seq here. In addition, a number of other assays have been used to validate these interactions both *in vivo* and *in vitro*. We will do the CLIP assay in future experiments to see whether there are other RNAs that also bind with these enzymes, and perform in depth investigation for newly identified RNAs if there are any.